# CAN WE USE GRADIENT NORM AS A MEASURE OF GENERALIZATION ERROR FOR MODEL SELECTION IN PRACTICE?

## ABSTRACT

The recent theoretical investigation (Li et al., 2020) on the upper bound of generalization error of deep neural networks (DNNs) demonstrates the potential of using the gradient norm as a measure that complements validation accuracy for model selection in practice. In this work, we carry out empirical studies using several commonly-used neural network architectures and benchmark datasets to understand the effectiveness and efficiency of using gradient norm as the model selection criterion, especially in the settings of hyper-parameter optimization. While strong correlations between the generalization error and the gradient norm measures have been observed, we find the computation of gradient norm is time consuming due to the high gradient complexity. To balance the trade-off between efficiency and effectiveness, we propose to use an accelerated approximation (Goodfellow, 2015) of gradient norm that only computes the loss gradient in the Fully-Connected Layer (FC Layer) of DNNs with significantly reduced computation cost (200~20,000 times faster). Our empirical studies clearly find that the use of approximated gradient norm, as one of the hyper-parameter search objectives, can select the models with lower generalization error, but the efficiency is still low (marginal accuracy improvement but with high computation overhead). Our results also show that the bandit-based or population-based algorithms, such as BOHB, perform poorer with gradient norm objectives, since the correlation between gradient norm and generalization error is not always consistent across phases of the training process. Finally, gradient norm also fails to predict the generalization performance of models based on different architectures, in comparison with state of the art algorithms and metrics.

## 1 INTRODUCTION

Generalization performance of deep learning through stochastic gradient descent-based optimization has been widely studied in recent work (Li et al., 2020; Chatterjee, 2020; Negrea et al., 2019; Thomas et al., 2019; Zhu et al., 2019; Hu et al., 2019; Zhang et al., 2016; Mou et al., 2017). These studies initialize the analysis from the learning dynamics' perspectives and then extend to the upper bound estimation of generalization errors (Mou et al., 2017). More recently, researchers have shifted their focuses onto providing some theoretical or empirical measures on generalization performance (Li et al., 2020; Negrea et al., 2019; Thomas et al., 2019; Cao & Gu, 2019; He et al., 2019; Frei et al., 2019) with respect to deep architectures, hyper-parameters, data distributions, learning dynamics and so on. This work studies the use of generalization performance measures (Li et al., 2020; Negrea et al., 2019) for model selection purposes (Cawley & Talbot, 2010).

Prior to the endeavor of deep learning, the generalization gap has been used as a straightforward measure. Given a set of $N$ training samples $\{x_1, x_2, x_3, \dots x_N\}$, a deep learning model $\theta$ and a loss function $\mathcal{L}(\theta; x)$ based on the sample $x$, the generalization gap $\mathcal{G}$ is defined as

$$\mathcal{G}(\theta) = \mathop{\mathbb{E}}_{x \sim \mathcal{X}} \mathcal{L}(\theta; x) - \frac{1}{N} \sum_{i=1}^{N} \mathcal{L}(\theta; x_i), \tag{1}$$

where $\mathcal{X}$ is defined as the distribution of data, and $\mathbb{E}_{x \sim \mathcal{X}} \mathcal{L}(\theta; x)$ refers to the expected loss. To quantify the generalization gap, validation or testing samples have been frequently used to measure the expected loss (Kohavi et al., 1995). Such that given a validation/testing dataset with $M$ samples

$\{y_1, y_2, y_3, \ldots, y_M\}$, the empirical generalization gap $\widehat{\mathcal{G}}_M$ is estimated as

$$\widehat{\mathcal{G}}_M(\theta) = \frac{1}{M} \sum_{i=1}^{M} \mathcal{L}(\theta; y_i) - \frac{1}{N} \sum_{i=1}^{N} \mathcal{L}(\theta; x_i), \tag{2}$$

and $\lim_{M \to \infty} \widehat{\mathcal{G}}_M = \mathcal{G}$. However, due to the limited amount of samples, the accurate estimation of generalization gap is not always available (Cawley & Talbot, 2010). It has been evidenced that performance tuning based on validation set frequently causes overfitting to the validation set (Recht et al., 2018) in deep learning settings.

Rather than the use of empirical generalization gap, some advanced measure has been proposed (Li et al., 2020; Negrea et al., 2019; Thomas et al., 2019) to provide data-dependent characterization on the generalization performance for deep learning. For example, Thomas et al. (2019) derived the Takeuchi information criterion (TIC) using the Hessian and covariance matrices of loss gradients with low-complexity approximation. They then propose to use TIC as an empirical metric that correlates to the generalization gap. As the calculation of TIC relies on the use of validation set, TIC for deep learning is a posterior measure. Further, Negrea et al. (2019) improved mutual information bounds for Stochastic Gradient Langevin Dynamics via some data-dependent measure. More specifically, the squared norm of gradients have been used as the data-dependent priors to tightly bound the generalization gap through measuring the flatness of empirical risk surface.

More recently, the squared norm of gradients over the learning dynamics has been studied as the measure to upper-bound the generalization gap (Li et al., 2020; Negrea et al., 2019). All above methods connect the generalization gap of deep learning to the gradients and Hessians of loss functions. While Thomas et al. (2019) is a posterior measure relying on the validation datasets, the two studies (Li et al., 2020; Negrea et al., 2019) provide "prior" measure that uses training datasets. More specifically, Li et al. (2020) proves that the generalization error is upper bounded by the empirical squared gradient norm along the optimization path. Formally, they show that, given a model trained with $n$ samples in dataset $\mathcal{X} = \{x_1, x_2 \ldots x_n\}$ for a total of $T$ iterations using a $C$-bounded loss function is used, the theoretical generalization gap of the DNN $\theta_T$ is bounded as follows,

$$\mathcal{G}(\theta_T) \leq \frac{2\sqrt{2}C}{n} \sqrt{\mathop{\mathbb{E}}_{\mathcal{X}} \left[ \sum_{t=1}^{T} \frac{\gamma_t^2}{\sigma_t^2} \mathbf{g}_e(t) \right]}, \tag{3}$$

where $\mathbf{g}_e(t) = \mathbb{E}_{\theta_{t-1}}[\frac{1}{n} \sum_{i=1}^{n} \|\nabla \mathcal{L}(\theta_{t-1}, x_i)\|_2^2]$ is the empirical squared gradient norm in the $t^{th}$ iteration, $\gamma_t$ is the learning rate and $\sigma_t$ indicates the standard deviation of Gaussian noise in the stochastic process. Their result directly shows that empirical squared gradient norm along the optimization path is a good indicator of a model's generalization ability. Therefore, we are interested in using this signal as a criterion in model selection, so as to find an optimal set of hyper-parameters.

**Our Contributions.** In this work, we follow the theoretical investigation in (Li et al., 2020) and try to use the *squared gradient norms (GN) over the optimization path* as a data-dependent generalization performance measure for DNN model selection. In an empirical manner, the proposed metric GN here should be able to measure the generalization gap of the DNN model $\theta_T$, which has been trained with $T$ iterations and $N$ training samples, as follows

$$\text{GN}(\theta_T) = \sum_{t=1}^{T} \left[ \frac{1}{N} \sum_{i=1}^{N} \|\nabla \mathcal{L}(\theta_t; x_i)\|_2^2 \right], \tag{4}$$

where $\nabla \mathcal{L}(\theta_t; x_i)$ refers to the loss gradient using the model of the $t^{th}$ iteration and the $i^{th}$ training sample in $\{x_1, x_2, \ldots, x_N\}$. Based on the data-dependent measure GN, our work make three pieces of significant technical contributions as follows

**(1)** *Approximated Gradient Norm (AGN) as an accelerated and low-complexity approximation to GN.* Despite the fact that GN can measure the generalization error of a DNN model, the time consumption for GN is high, due to the complexity of gradient estimation using every training sample in every iteration. To lower the computational complexity, we propose *Approximated Gradient Norm* (AGN) that only uses the summation of loss gradients of the Fully-Connected (FC) Layer in DNN by the end of every epoch (rather than every iteration) as an approximation of GN. Furthermore, the calculation of FC Layer gradient per sample could be further accelerated by Goodfellow

(2015) with extremely low costs. Our empirical evaluation finds that, over various DNN models trained with different hyper-parameters, the metrics $GN(\theta_T)$ and $AGN(\theta_T)$ behave identically with respect to empirical generalization gap $\widehat{\mathcal{G}}_M(\theta_T)$. This approximation makes it feasible to carry out experiments that evaluate the effectiveness of GN for model selection.

**(2)** To validate the correlations between generalization performance and $AGN(\theta_T)$, we carry out extensive experiments using various deep neural networks, such as Multi-Layer Perception (MLP), LeNet (LeCun et al., 1998), and ResNet (He et al., 2016), on top of benchmark datasets including MNIST (LeCun et al., 2010), Fashion-MNIST (Xiao et al., 2017), SVHN (Netzer et al., 2011), and CIFAR (Krizhevsky et al., 2009). **Observation 1**: It has been observed that, when the models are *well-fitted* (i.e., with low training loss or high training accuracy), $AGN(\theta_T)$ well corresponds with the empirical generalization gap $\widehat{\mathcal{G}}_M(\theta_T)$ while models with lower/higher $AGN(\theta_T)$ are consistently with lower/higher $\widehat{\mathcal{G}}_M(\theta_T)$ and better/poorer generalization performance. On the other hand, when models are not well-fitted or namely *under-fitted*, either due to the use of inappropriate hyper-parameters or a training process not converged (e.g., $T$ is small), the correlations between $\widehat{\mathcal{G}}_M(\theta_T)$ and $AGN(\theta_T)$ are not always consistent while the direction of correlation somtimes is even opposite to the theoretical investigation in Li et al. (2020) and Eq. (3).

**(3)** To understand the effectiveness and efficiency of using $AGN(\theta_T)$ for model selection, we extend our experiments to use $AGN(\theta_T)$ as an objective for hyper-parameter selection, under both *black-box optimization* (Escalante et al., 2009; Loshchilov & Hutter, 2016) and *bandit-based search* (Li et al., 2017; Falkner et al., 2018) settings. **Observation 2**: We find that, through the combination with training or validation loss, $AGN(\theta_T)$ can help black-box optimization algorithms, such as particle swarm (PSO) (Escalante et al., 2009) or covariance matrix adaption based evolving strategies (CMA-ES) (Loshchilov & Hutter, 2016), search the models (hyper-parameters) with equivalent or marginally better performance than using validation loss/accuracy as the search objective. The training procedure can somehow avoid the potential overfitting to the validation set (Recht et al., 2018). However the use of $AGN(\theta_T)$ requests more prior knowledge (additional parameters) to balance the weights of training/validation loss and $AGN(\theta_T)$ in the combined objective during the search, which might be sensitive for model selection. **Observation 3**: Further, our research finds that *bandit-based* search algorithms, such as Bayesian optimization over HyperBand (BOHB) (Falkner et al., 2018), cannot work well with gradient norms. Because BOHB selects and drops models during the learning process with respect to the intermediate measures on performance, while within an end-to-end learning process $AGN(\theta_t)$ (for $\theta_1, \theta_2, \theta_3 \ldots, \theta_T$) cannot always provide consistent measures on the generalization performance. **Observation 4**: In addition to hyper-parameter selection for deep learning, our experiment finally finds that $AGN(\theta_T)$ fails to predict the the generalization performance of the models based on different deep architectures, in comparison to some state of the art algorithms and metrics (Jiang et al., 2019; Nagarajan & Kolter, 2019).

We believe the most relevant studies to this work are those done by Li et al. (2020); Negrea et al. (2019); Thomas et al. (2019) (that measure the generalization performance of DNN using derivatives, such as gradients and Hessian matrices, of the loss), and the contribution made by our work is unique. Compared to Thomas et al. (2019), the measure AGN studied here is also a data-dependent metric to characterize the generalization performance. AGN uses gradients and avoids the use of validation sets, while Thomas et al. (2019) is based on Hessian matrices and relies on validation data. Compared to Li et al. (2020); Negrea et al. (2019), our work studies the feasibility of using gradient norms over learning process for model selection in empirical settings, where we first provide AGN as a low-complexity implementation to accelerate the computation of gradient norms, then we conduct extensive results to demonstrate the pros and cons of using gradient norms or AGN for model selection in two major hyper-parameter search settings. Our result demonstrates the potentials but also limitations in the adoption of current theoretical results (Li et al., 2020; Negrea et al., 2019) as an objective for model selection.

## 2 GRADIENT NORM AND ITS APPROXIMATION

Given a DNN model $\theta_T$ that has been trained with $T$ iterations, the straightforward way to compute $GN(\theta_T)$ is to first collect and store the weights of the model for every iteration during the training process (i.e., $\theta_1, \theta_2, \ldots, \theta_T$), then compute the gradient for every training sample ($N$ in total) on every model ($T$ in total) through backpropagation (BP) over the whole DNN for $N \times T$ times. In

practice, such computation cost is far too high. In this section, we present $\text{AGN}(\theta_T)$ as a low-complexity approximation to $\text{GN}(\theta_T)$ and then discuss the approximation performance.

## 2.1 AGN: APPROXIMATED GRADIENT NORM

We propose $\text{AGN}(\theta_T)$ as a low-complexity approximation to accelerate the computation of $\text{GN}(\theta_T)$ using acceleration strategies as follows.

**Depth-wise Acceleration.** While DNNs frequently composes hundreds of layers, $\text{AGN}(\theta_T)$ approximates to $\text{GN}(\theta_T)$ using the gradients of last Fully-Connected (FC) Layer of DNNs.

**Sample-wise Acceleration.** While the computation of $\text{GN}(\theta_T)$ needs to run time-consuming BP for every training sample, $\text{AGN}(\theta_T)$ uses the low-complexity per-sample gradient estimation algorithm (Goodfellow, 2015) to compute the gradients of FC layers without BP.

**Epoch-wise Acceleration** While the computation of $\text{GN}(\theta_T)$ aggregates the gradient norms for every iteration, $\text{AGN}(\theta_T)$ approximates to $\text{GN}(\theta_T)$ via the summation of squared norms of gradients collected by the end of every epoch. Given the batch size $B$ for every iteration, each epoch consists of $\frac{N}{B}$ gradient descent iterations, the learning process with $T$ iterations takes $\frac{TB}{N}$ epochs for completion, and $\theta_{\frac{\tau N}{B}}$ for $\tau = 1, 2, \ldots, \frac{TB}{N}$ refer to the model obtained by the end of every epoch.

In this way, we propose to compute $\text{AGN}(\theta_T)$ as follows

$$\text{AGN}(\theta_T) = \sum_{\tau=1}^{TB/N} \left[ \frac{1}{N} \sum_{i=1}^{N} \left\| \nabla \mathcal{L}^{\text{FC}}(\theta_{\frac{\tau N}{B}}; x_i) \right\|_2^2 \right],$$

(5)

where $\nabla \mathcal{L}^{\text{FC}}(\theta; x)$ refers to the gradient of FC layer based on the model $\theta$ and the sample $x$ and is accelerated by using the algorithm proposed by Goodfellow (2015).

## 2.2 APPROXIMATION PERFORMANCE

We discuss the approximation performance of $\text{AGN}(\theta_T)$ to $\text{GN}(\theta_T)$ from efficiency and effectiveness perspectives, in comparison with validation accuracy, as ways of generalization performance measurements.

**Effectiveness Comparison.** We find that $\text{AGN}(\theta_T)$ preserves the effectiveness of $\text{GN}(\theta_T)$ as a generalization performance indicator. In Figure 1, we illustrate the comparison

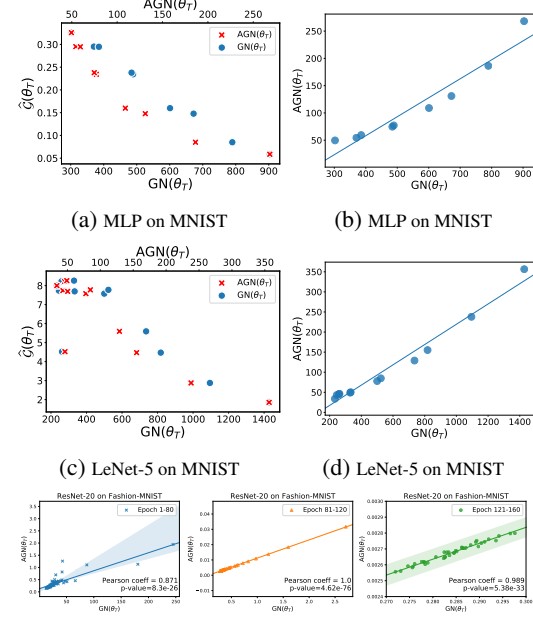

(a) MLP on MNIST      (b) MLP on MNIST

(c) LeNet-5 on MNIST      (d) LeNet-5 on MNIST

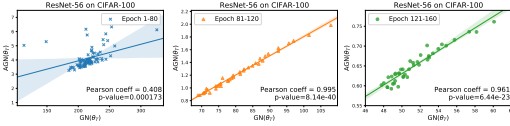

(e) ResNet-20 on Fashion-MNIST(validating the positive correlation between $\text{GN}(\theta_T)$ and $\text{AGN}(\theta_T)$ in larger neural networks).

(f) ResNet-56 on CIFAR-100 (validating the positive correlation between $\text{GN}(\theta_T)$ and $\text{AGN}(\theta_T)$ in larger neural networks).

Figure 1: Comparison between $\text{GN}(\theta_T)$ and $\text{AGN}(\theta_T)$ in terms of effectiveness to reflect generalization error. In Fig. 1a to Fig. 1d, we pick 9 sets of random hyper-parameters for MLP and 12 sets for LeNet-5. We then train the model for 40 epochs and plot their results. For the two architectures we have tested on MNIST, both $\text{AGN}(\theta_T)$ and $\text{GN}(\theta_T)$ exhibit identical trends when generalization error is plotted against them respectively. Strong positive correlation is observed between GN and AGN in each setting as well, with Pearson correlation coefficients 0.962 for MLP and 0.987 for LeNet-5. We carry out additional experiments using deeper neural network models in Fig. 1e and Fig. 1f in order to verify the existence of strong positive correlation between $\text{AGN}(\theta_T)$ and $\text{GN}(\theta_T)$. We follow a piecewise learning rate decay policy in these experiments. Thus, there are three plots for each ResNet setting.

between $\text{AGN}(\theta_T)$ and $\text{GN}(\theta_T)$ using 9 Multi-Layer Preceptors (MLP) models and 12 LeNet-5 models trained using 9 and 12 sets of random hyper-parameters on MNIST datasets. For every model here, we train the model with 40 epoch, measure the generalization gap using the validation set, and estimate $\text{GN}(\theta_T)$ and $\text{AGN}(\theta_T)$ respectively. In Figure 1(a) and (c), we plot $\text{GN}(\theta_T)$ and $\text{AGN}(\theta_T)$ of these models against their generalization gaps. It shows that, despite the different scales of the two measures, the trends of $\text{GN}(\theta_T)$ and $\text{AGN}(\theta_T)$ behave identically with respect

to the generalization gaps. Further, we correlate $\text{GN}(\theta_T)$ and $\text{AGN}(\theta_T)$ for every model and demonstrate the correlations in Figure 1 (b) and (c). We carry out additional experiments on larger neural networks. The results in Fig. 1e and Fig. 1f also validate that the correlations between $\text{GN}(\theta_T)$ and $\text{AGN}(\theta_T)$ are strong, significant, and consistent.

**Efficiency Comparison.** We find that the computation of $\text{AGN}(\theta_T)$ is much faster than $\text{GN}(\theta_T)$ while it still consumes significantly more time than using validation set for the measurement of generalization performance. In Figure 2, we plot the time consumption per epoch of the three generalization performance measurements, i.e., $\text{AGN}(\theta_T)$,

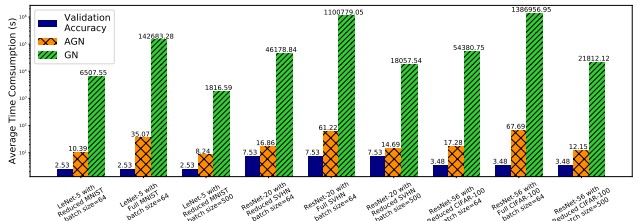

Figure 2: Comparisons in computation time for the three generalization performance measurements.

$\text{GN}(\theta_T)$, and validation accuracy, using LeNet-5, ResNet-20, and ResNet-56 on (parts of) MNIST, SVHN, and CIFAR-100 datasets. To simulate the settings of realistic hyper-parameter search, we collect time consumption of the three measurements using full and reduced (20%) datasets separately. The comparison results show that $\text{AGN}(\theta_T)$ is 220x$\sim$20,489x faster than $\text{GN}(\theta_T)$ while it still consumes 95%$\sim$ 1845% more time than using validation accuracy. Note that, for the comparison, we originally collect the time consumption of $\text{GN}(\theta_T)$ for a single iteration, and then rescale the figures to one epoch.

In summary, we conclude that $\text{AGN}(\theta_T)$ is a low-complexity but tight approximation to $\text{GN}(\theta_T)$ for generalization performance measurement.

# 3 USING AGN AS A GENERALIZATION PERFORMANCE INDICATOR

To understand the correlations between $\text{AGN}(\theta_T)$ and generalization performance, we propose to measure $\widehat{\mathcal{G}}_M(\theta_T)$ and $\text{AGN}(\theta_T)$ of a wide range of DNN models using ResNet-20, ResNet-56, ResNet-110, and DenseNet100x24 (Huang et al., 2017) based on CIFAR-10 datasets. For each architecture, we train 32 DNN models, each of which is with a random set of hyper-parameters and the same number of epochs. Specifically, we hope to discuss the correlations in two scenarios – the model is *well-fitted* or *under-fitted*. With sufficient number of training iterations and appropriate settings of hyper-parameters, the models are usually trained to *well-fit* the training datasets with low training loss and high training accuracy. However, due to the lack of convergence or inappropriate setting, some models are *under-fitted* even using a large number of iterations.

**Observation 1** *The correlations between* $\text{AGN}(\theta_T)$ *and* $\widehat{\mathcal{G}}_M(\theta_T)$ *are not consistent from models to models.* $\text{AGN}(\theta_T)$ *is positively correlated with* $\widehat{\mathcal{G}}_M(\theta_T)$ *when models are well-fitted. Otherwise, an opposite trend is observed for underfitted models.* Fig. 3 illustrates that the inconsistent correlations between $\text{AGN}(\theta_T)$ and $\widehat{\mathcal{G}}_M(\theta_T)$ appear in all these architectures. For each architecture, 32 random hyperparameter configurations were chosen to train 32 models, and we obtain each model's $\text{AGN}(\theta_T)$ at the end of 160 epochs as well as its generalization error, plotted in Fig. 3a. We observe that the scatters exhibit a trend that looks like an inverted checkmark. This shape indicates that the correlation between $\text{AGN}(\theta_T)$ and generalization error behave differently based on model con-

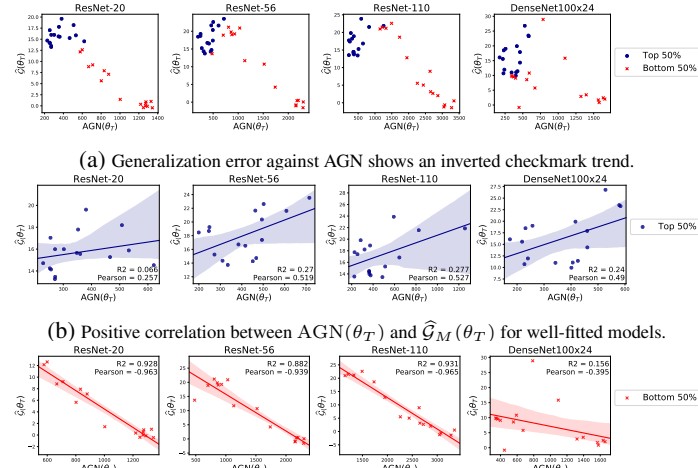

(a) Generalization error against AGN shows an inverted checkmark trend.

(b) Positive correlation between $\text{AGN}(\theta_T)$ and $\widehat{\mathcal{G}}_M(\theta_T)$ for well-fitted models.

(c) Negative correlation between $\text{AGN}(\theta_T)$ and $\widehat{\mathcal{G}}_M(\theta_T)$ for under-fitted models.

Figure 3: Inconsistent correlation between empirical squared gradient norm and generalization error depending on how well a model is trained.

vergence. Due to this inconsistent behavior, it is not always possible to use $\mathrm{AGN}(\theta_T)$ as an indicator of generalization performance unless the models are well-fitted.

*Positive correlation between* $\mathrm{AGN}(\theta_T)$ *and* $\widehat{\mathcal{G}}_M(\theta_T)$ *for well-fitted models* We identify the models with top 50% training accuracy. These models have fitted the training set with at least 88% accuracy and hence they are well-fitted models. These models show positive correlation between $\mathrm{AGN}(\theta_T)$ and $\widehat{\mathcal{G}}_M(\theta_T)$, as shown in Fig. 3b. It is feasible to use $\mathrm{AGN}(\theta_T)$ as an alternative generalization performance indicator, in addition to the validation loss/accuracy, under these settings. This observation coincides with the theoretical findings in Li et al. (2020) and Eq. (3). However, the p-values for Pearson correlation in Fig.3(b) are 0.336, 0.039, 0.036, and 0.053 respectively. These values overall indicate weak positive correlation between generalization gap and $\mathrm{AGN}(\theta_T)$ as three of them are close to or lower than 0.05. This weak correlation is most likely a result of that $\mathrm{GN}(\theta_T)$ does not work very well as a metric in practice, since we have already shown the experimental results that indicate good approximation of $\mathrm{AGN}(\theta_T)$ to $\mathrm{GN}(\theta_T)$ in Fig. 1.

*Negative correlation between* $\mathrm{AGN}(\theta_T)$ *and* $\widehat{\mathcal{G}}_M(\theta_T)$ *for under-fitted models* In contrast, we also identify the models with bottom 50% training accuracy. Such models are under-fitted models due to inappropriate settings of hyper-parameters or lack of convergence in the training process. Fig. 3c shows that $\mathrm{AGN}(\theta_T)$ is negatively correlated with $\widehat{\mathcal{G}}_M(\theta_T)$ for under-fitted models. We however do not consider this observation conflicts with the theoretical findings in Li et al. (2020) or Eq. (3), because Li et al. (2020) actually studied the learning dynamics of DNN in asymptotic settings, where authors derived the upper bound of generalization error using gradient norms when $T \to \infty$ (the training procedure has been well converged and models are well-fitted). Nonetheless, it still seems to be surprising that there is a negative correlation between $\mathrm{AGN}(\theta_T)$ and $\widehat{\mathcal{G}}_M(\theta_T)$. We offer an intuitive explanation here. The trend is that gradient norm gradually decreases as training proceeds while generalization gap increases (also reported in Li et al. (2020)). Across models with different hyper-parameters, these two quantities could possibly change their magnitude at very different speed. A model that remains under-fitted throughout training has low generalization gap (it makes almost random guesses on both training and test sets), but the training samples are likely to consistently produce large gradients. As a result, under-fitted models with smaller generalization gaps have larger $\mathrm{AGN}(\theta_T)$, contributing to the negative correlation shown in Fig. 3c.

## 4 USING AGN FOR MODEL SELECTION

In this section, we investigate the feasibility of using $\mathrm{AGN}(\theta_T)$ as a generalization performance indicator for model selection. We consider two possible applications of $\mathrm{AGN}(\theta_T)$, where the first one is to incorporate $\mathrm{AGN}(\theta_T)$ as an objective for hyper-parameter search for a fixed architecture, and the second one is using $\mathrm{AGN}(\theta_T)$ to predict cross-architectural models' generalization error.

### 4.1 USE CASE 1: HYPER-PARAMETER SEARCH FOR DEEP NEURAL NETWORKS

To address the fitness issues from Observation 1, we propose to include the training loss and/or validation loss as the search objectives as follows

$$\mathrm{Objective}_{\mathrm{AGN+Train}}(\theta_T) = \alpha \cdot \mathrm{AGN}(\theta_T) + \frac{1}{N}\sum_{i=1}^{N}\mathcal{L}(\theta_T; x_i),$$

$$\mathrm{Objective}_{\mathrm{AGN+Val}}(\theta_T) = \alpha \cdot \mathrm{AGN}(\theta_T) + \frac{1}{M}\sum_{i=1}^{M}\mathcal{L}(\theta_T; y_i),$$

(6)

where $\{x_1, x_2, \ldots, x_N\}$ and $\{y_1, y_2, \ldots, y_M\}$ refer to the training and validation sets respectively, and $\alpha$ is a weight to balance the training/validation loss and $\mathrm{AGN}(\theta_T)$.

We test the effectiveness of the above two search objectives by conducting hyper-parameter search using ResNet-20 and ResNet-56 on Fashion-MNIST, SVHN, and CIFAR datasets respectively. The results of Fashion-MNIST and SVHN are found in the appendix. On top of the two objectives in Eq. (6), we try multiple settings of $\alpha$ and adopt two commonly-used black-box optimization algorithms, including CMA-ES (Loshchilov & Hutter, 2016) and PSO (Escalante et al., 2009), for hyper-parameter search with a computation budget of 5,120 GPU×epochs, where we take a reduced training set with 20% training samples (Cubuk et al., 2019) to accelerate the search procedure. The search space is with Batch Size [32, 1000], Learning Rate [$1 \times 10^{-7}$, 0.50], Momentum [0, 0.99], and Weight Decay [$5 \times 10^{-7}$, 0.05], while initial settings are Batch Size = 100, Learning Rate = 0.01, Momentum = 0.6, and Weight Decay = 0.0005.

Table 1: Results for CIFAR-10 and CIFAR-100 using CMA-ES and PSO.

| Method | CIFAR-10 | | CIFAR-100 | |
| --- | --- | --- | --- | --- |
| | ResNet-20 | ResNet-56 | ResNet-20 | ResNet-56 |
| Default | 91.70±0.16 | 92.97±0.18 | 66.73±0.56 | 70.37±0.46 |
| CMA-ES(Val) | 91.86±0.13 | 92.74±0.20 | **68.28±0.46** | 67.08±0.68 |
| PSO(Val) | **92.26±0.09** | 93.15±0.09 | 67.85±0.25 | 72.11±0.39 |
| CMA-ES(AGN+Train,$\alpha$=0.05) | 92.22±0.09 | **93.97±0.28** | 64.90±0.20 | 71.30±0.17 |
| CMA-ES(AGN+Train,$\alpha$=0.01) | 90.46±0.09 | 91.59±0.19 | 64.62±0.04 | 72.37±0.12 |
| CMA-ES(AGN+Train,$\alpha$=0.005) | 90.36±0.04 | 93.75±0.18 | 66.37±0.38 | 72.18±0.21 |
| CMA-ES(AGN+Val,$\alpha$=0.1) | 91.94±0.07 | 93.09±0.04 | 64.64±0.50 | 69.14±0.17 |
| CMA-ES(AGN+Val,$\alpha$=0.05) | 91.56±0.14 | 92.92±0.15 | 66.65±0.46 | 70.04±0.11 |
| CMA-ES(AGN+Val,$\alpha$=0.005) | 92.08±0.08 | 93.77±0.06 | 66.31±0.11 | **72.49±0.46** |
| PSO(AGN+Train,$\alpha$=0.05) | 90.56±0.11 | 91.66±0.06 | 62.09±0.26 | 69.28±0.16 |
| PSO(AGN+Train,$\alpha$=0.01) | 91.51±0.04 | 91.32±0.10 | 67.46±0.18 | 67.55±0.52 |
| PSO(AGN+Train,$\alpha$=0.005) | 91.92±0.14 | 91.99±0.31 | 64.91±0.17 | 67.05±0.43 |
| PSO(AGN+Val,$\alpha$=0.1) | 90.38±0.03 | 92.55±0.06 | 66.57±0.16 | 67.92±0.39 |
| PSO(AGN+Val,$\alpha$=0.05) | 91.67±0.18 | 91.76±0.05 | 65.53±0.35 | 71.42±0.13 |
| PSO(AGN+Val,$\alpha$=0.005) | 90.66±0.08 | 93.76±0.17 | 67.02±0.12 | 71.51±0.19 |

**Default hyper-parameter setting** For every pair of dataset and the DNN architecture, there exists a "default" set of hyper-parameters that has been frequently used/suggested by practitioners, or published in their official releases. For example, the hyper-parameters including Batch Size = 128, Learning Rate = 0.1, Momentum = 0.9, and Weight Decay = $10^{-4}$ have been frequently used to train ResNet-20 on CIFAR-10. In the tables following, "Default" refers to models trained with the default hyper-parameter configuration.

**Observation 2.** *When combining* $\mathrm{AGN}(\theta_T)$ *with training or validation loss as search objectives, black-box optimization algorithms can search the hyper-parameters with performance marginally better than using validation accuracy only, whilst models can avoid overfitting to the validation set (Recht et al., 2018). However, such practice requests prior knowledge (i.e., $\alpha$ in Eq. (6)) to balance the two factors in the search objective.* Table 1 and Table 4 (in appendix) present the results of comparisons, where we include the testing accuracy of the models trained using full training set and the hyper-parameters found by black-box optimization algorithms. For every set of searched hyper-parameters, the model has been trained and tested three times with error bars estimated. For both CMA-ES and PSO algorithms, the two objectives can help search the models with significantly better or worse testing accuracy than the one selected by using validation accuracy as objectives.

Furthermore, the use of these two objectives can fix the issues of *overfitting to the validation set* for model selection. For example, the result of ResNet-20 on Fashion-MNIST using validation set in Table 4 has a lower test set accuracy than the model trained with default hyper-parameters. This phenomenon is likely due to overfitting the validation set since ResNet-20 has a rather large capacity to learn and Fashion-MNIST is a relatively easy dataset. In contrast, AGN+Train or AGN+Val leads to slightly better result than the default model, significantly outperforming using validation loss. The final model selection highly depends on the choice of $\alpha$. We however do not know such prior in advance, while trends of performance over $\alpha$ are not consistent or obvious. In this way, we can conclude that there exists potentials of using $\mathrm{AGN}(\theta_T)$ as part of search objectives for black-box hyper-parameter optimization when the prior knowledge on the settings of $\alpha$ is known. Otherwise, the use of $\mathrm{AGN}(\theta_T)$ might even hurt the hyper-parameter search, no matter whether it is combined with training or validation loss.

**Observation 3.** $\mathrm{AGN}(\theta_T)$ *sometimes cannot work well with bandit-based hyper-parameter search, as the correlation between* $\mathrm{AGN}(\theta_T)$ *and generalization performance is not always consistent during the training process from under-fitted to well-fitted status.* We carry out the similar hyper-parameter search experiments using BOHB algorithm under the same settings (search space, initial values, and computation budget). For fair comparison, BOHB normalizes $\mathrm{AGN}(\theta_t)$ in the objectives (and the setting of $\alpha$ is slightly different from the black-box optimization settings), since BOHB has to compare $\mathrm{AGN}(\theta_t)$ of models obtained from different iterations $t$ of the training process while the scale of $\mathrm{AGN}(\theta_t)$ varies significantly.

Table 2: Results for ResNet-20 on Fashion-MNIST, CIFAR-10, and CIFAR-100 using BOHB.

| | Fashion-MNIST | CIFAR-10 | CIFAR-100 |
|---|---|---|---|
| Default | 93.34±0.07 | **91.70±0.16** | 66.73±0.56 |
| BOHB (Val) | 92.58±0.64 | 90.46±0.65 | 66.02±0.32 |
| BOHB (AGN+Train, $\alpha$=0.9) | 91.87±0.51 | 89.73±0.17 | 66.53±0.37 |
| BOHB (AGN+Train, $\alpha$=0.5) | 89.25±2.00 | 85.02±0.19 | 62.19±0.44 |
| BOHB (AGN+Train, $\alpha$=0.1) | **93.41±0.17** | 91.18±0.09 | 59.67±0.10 |
| BOHB (AGN+Val, $\alpha$=0.9) | 93.01±0.20 | 85.88±0.87 | 63.30±0.41 |
| BOHB (AGN+Val, $\alpha$=0.5) | 82.53±9.58 | 89.07±0.47 | **66.73±0.33** |
| BOHB (AGN+Val, $\alpha$=0.1) | 93.27±0.06 | 90.61±0.16 | 64.05±0.09 |

In Table 2, we present the testing accuracy of the models that are trained using the hyper-parameters searched by BOHB with the objectives. The results show that, though it is able to outperform the default one (such as ResNet-20 on Fashion-MNIST using AGN+Train, $\alpha = 0.1$), using $\mathrm{AGN}(\theta_t)$ within BOHB cannot produce better results than the default setting in most cases. Since the correlation between $\mathrm{AGN}(\theta_t)$ and $\widehat{\mathcal{G}}_M(\theta_t)$ is not always consistent over the optimization path, AGN gives BOHB inaccurate feedback on model generalization performance in early stage of training. Consequently, BOHB tends to early terminate some good models and continue training the inferior ones. This results in deteriorated model performance through the bandit-based search.

## 4.2 USE CASE 2: CROSS-ARCHITECTURAL GENERALIZATION ERROR PREDICTION

We evaluate a series of complexity measures to test their effectiveness of predicting generalization error using the public dataset provided in a competition at NeurIPS 2020 named "Predicting Generalization in Deep Learning" (Jiang et al.). The descriptions of experimental setup and public dataset are available in the official document of competition details. Briefly, we use 150 trained VGG-like architectures (along their datasets used during training, namely SVHN and CIFAR-10) as datum and predict their generalization error with the listed theoretical complexity measures in Table 3. The predictions are evaluated using conditional mutual information. The possible scores are between 0 and 100. A higher score indicates that the complexity measures is able to more accurately and consistently predict the generalization error. The results in Table 3 show that although $\mathrm{AGN}(\theta_T)$ is able to outperform some baseline theoretical complexity measures in predicting generalization error, its effectiveness is largely limited. The ineffectiveness of using $\mathrm{AGN}(\theta_T)$

Table 3: Cross-architectural generalization error prediction scores of complexity measures. VC dimension (Vapnik & Chervonenkis, 2015), Jacobian Norm with respect to intermediate layers (Jiang et al.), Distance of trained weights from initialization ($\theta_0$) (Nagarajan & Kolter, 2019), and Sharpness of local minima (Jiang et al., 2019)

| Method | Score |
|---|---|
| VC Dimension | 0.019598841 |
| Jacobian Norm | 2.061258975 |
| **AGN** | 3.955067687 |
| Distance to $\theta_0$ | 4.921479998 |
| Sharpness | 10.66711408 |

to perform cross-architectural generalization error prediction implies it is also ineffective to use $\mathrm{GN}(\theta_T)$ in practice since we have shown that $\mathrm{AGN}(\theta_T)$ approximates $\mathrm{GN}(\theta_T)$ very well.

## 5 CONCLUSION

In this paper, we have studied the feasibility of using $\mathrm{AGN}(\theta_T)$, an approximated form of squared norms of loss gradients over optimization path (Li et al., 2020), to measure generalization error in practice. We find the correlations between $\mathrm{AGN}(\theta_T)$ and $\widehat{\mathcal{G}}_M(\theta_T)$ are completely opposite for *under-fitted* and *well-fitted* models, where the positive correlations between the two variables found in well-fitted models coincide with the theorems by Li et al. (2020). Furthermore, the use of $\mathrm{AGN}(\theta_T)$ to complement the validation accuracy as the objectives can marginally improve the performance of hyper-parameter optimization, however the computational overhead caused by $\mathrm{AGN}(\theta_T)$ estimation and the lack of some prior knowledge makes such paradigm neither efficient nor effective. In the meanwhile, the same set of objectives does not bring any improvements for bandit-based algorithms such as BOHB, partially due to the inconsistent correlation between the objective and generalization performance throughout the training phase. As a result, using the gradient norms for model selection in practice remains challenging due to the high computation overhead (using the approximated version is up to 18 times slower than standard training) and limited effectiveness. Our experiments also show that $\mathrm{AGN}(\theta_T)$ cannot effectively predict generalization error given different architectures. In conclusion, we do not recommend using $\mathrm{GN}(\theta_T)$ or $\mathrm{AGN}(\theta_T)$ for model selection in practice.

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

# A  EXPERIMENTS

## A.1  EXPERIMENTAL SETUP

**Datasets and Data Augmentation**   We briefly introduce the benchmark datasets used in this paper and the data augmentation methods we applied when loading the data.

- Fashion-MNIST (Xiao et al., 2017) is a benchmark dataset similar to the popular MNIST dataset (LeCun et al., 2010). It has 70,000 grayscale images in the size of $28 \times 28$, dividing into 10 classes. The training set has 6,000 images from each class while the rest make up the test set. We normalize the input data as done in common practice.
- SVHN is a benchmark dataset composing house-number signs in street level images (Netzer et al., 2011). We use the cropped digits dataset for training and testing. We normalize the input data as done in common practice.
- The CIFAR-10 dataset has 6,000 examples of each of 10 classes and the CIFAR-100 dataset has 600 examples of each of 100 non-overlapping classes (Krizhevsky et al., 2009). We apply standard data augmentation the same way as the Pytorch official examples on CIFAR-10 and CIFAR-100 classification tasks. Specifically, we pad the input images by 4 pixels, and then randomly crop a sub-region of $32 \times 32$ and randomly do a horizontal flip. We normalize the input data as done in common practice.

**Reduced Dataset**   To reduce the expensive computation cost for training a model for multiple rounds, we train the model on the reduced dataset when searching for hyper-parameters, where only a random but fixed 20% subset of the standard dataset participates in the training. Training on the reduced dataset gives us a set of hyper-parameters, which is then used to train a separate model with the standard training set. The newly trained model is then validated on the standard test set. We report the final test performance in the main paper. The strategy of using a reduced dataset for hyper-parameter search is also used by Cubuk et al. (2019).

**ResNet Training Details**   For different architectures of ResNet, we train them for 160 epochs. For the default baseline model, we set the initial learning rate to be 0.1 and decay it by $1/10$ and $1/100$ at epoch 80 and 120 respectively. Momentum is set to be 0.9 and weight decay is 0.0001. The default batch size is 128. The loss function used for all experiments in the paper is cross entropy loss.

**Blackbox Optimization Setup**   The process of blackbox hyper-parameter optimization is split into 4 rounds. We train 8 models with different sets of hyper-parameters in parallel in one round. At the end of each round, a blackbox optimization algorithm is run based on the model score computed by using a validation set or using our metric. The algorithm generates the next 8 sets of hyper-parameters to search for the optimal ones. We use two blackbox optimization algorithms for hyper-parameter search, which are Covariance Matrix Adaptation - Evolution Strategy (CMA-ES) (Bergstra et al., 2011) and Particle Swarm Optimization (PSO) (Escalante et al., 2009).

**BOHB Hyper-parameter Search Setup**   We use BOHB (Falkner et al., 2018) to test how our proposed objectives work with bandit based training hyper-parameter optimization algorithms. For all ResNet architectures, the minimum budget is 40 and maximum is 160. We pick $\eta$ to be 2 so that the total number of training epochs using BOHB would be the same as that using blackbox optimization algorithms. This ensures fair comparison across the two types of hyper-parameter optimization algorithms.

## A.2  NOTES ON TIME CONSUMPTION ANALYSIS

**Interpretation of Figure. 2 in the Main Paper**   Each bar in the plot represents the total time consumption in one epoch, including time spent forwarding the data, updating the model parameter with backpropagation, and computing $\mathrm{GN}(\theta_T)$ or $\mathrm{AGN}(\theta_T)$ if needed. The results are obtained by running models multiple times on 1080Ti GPUs.

**GPU$\times$epochs**   We define 1 GPU$\times$epoch as iterating through the training set for one round on one GPU. Note that 1 GPU$\times$epoch corresponds to different amount of wall clock time depending on the

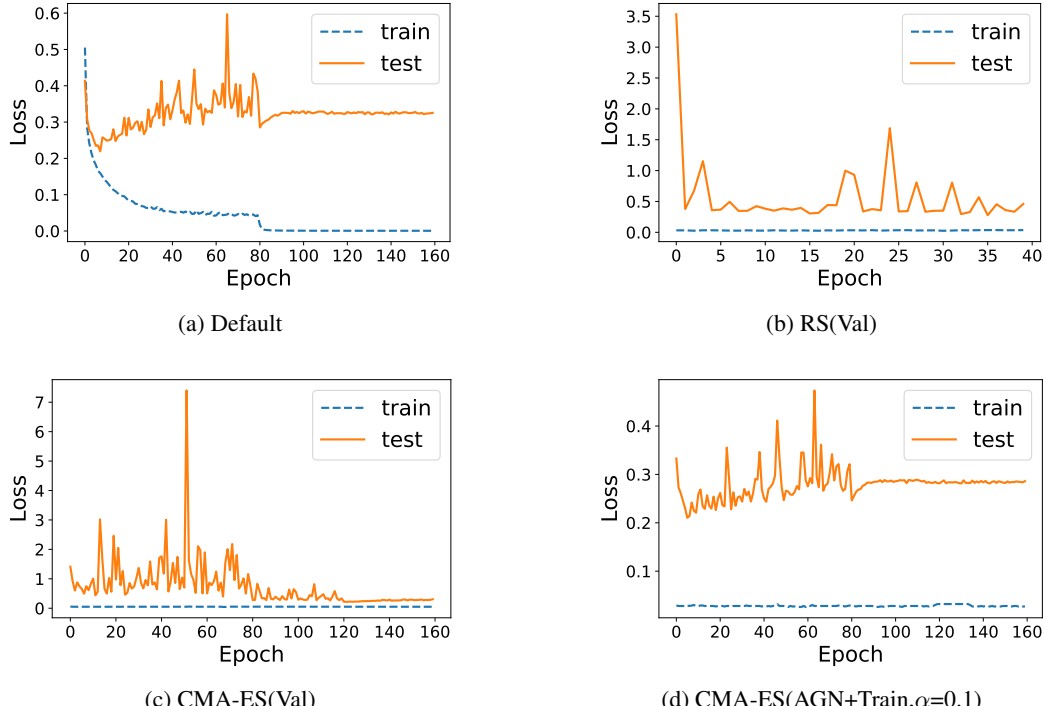

Figure 4: Training behavior of ResNet-20 on Fashion-MNIST with the default or learned hyper-parameters from various methods. "RS" refers to random search.

architecture, dataset as well as batch size. Figure 2 in the main paper shows some examples of the wall clock time that 1 GPU×epoch costs under those settings.

When running hyper-parameter search algorithms, we keep GPU×epochs as a constant variable to ensure each algorithm iterates through the training set the same number of times. As mentioned in the main paper, we set the total budget to be 5,120 GPU×epochs for all hyper-parameter search algorithms. Blackbox optimization algorithms used in this paper, namely CMA-ES and PSO, explores 32 sets of hyper-parameters whereas BOHB, as a more efficient search algorithm with early termination of poorly performing models, explores 36 sets of hyper-parameters in total.

**Specific time cost for hyper-parameter search.**    The time cost for hyper-parameter search using the reduced training set with validation accuracy as the criterion for ResNet-56 on CIFAR-10 is 856s per model on average. When AGN is used to complement validation accuracy, the average is 3,633s for one model.

### A.3   DISCUSSION ON TRAINING BEHAVIOR

Fig. 4 shows the training behavior of ResNet-20 on Fashion-MNIST with four different hyper-parameter settings. We analyze how the training and test losses vary under these settings and obtain the observations below.

**AGN Helps Reduce the Generalization Gap**    First, Fig. 4a indicates a gradually enlarging gap between the training loss and testing loss. This phenomenon is likely due to that we train ResNet-20 models for 160 epochs, which is easy to overfit the Fashion-MNIST dataset. As a result, there exists a generalization gap, where the training loss is very close to zero, but test loss is about 0.325. In contrast, when AGN and training loss are used as a combined objective in CMA-ES hyper-parameter optimization (Fig. 4d), the final generalization gap is 0.259, which is 20.3% lower than the generalization gap using the default hyper-parameter. This decrease in the magnitude of generalization

Table 4: Results for Fashion-MNIST and SVHN using CMA-ES and PSO.

| | Fashion-MNIST | | SVHN | |
|---|---|---|---|---|
| Method | ResNet-20 | ResNet-56 | ResNet-20 | ResNet-56 |
| Default | 93.34±0.07 | 93.21±0.24 | 95.67±0.02 | 96.11±0.14 |
| CMA-ES (Val) | 93.35±0.08 | 93.03±0.76 | **96.26±0.08** | 95.90±0.44 |
| PSO (Val) | 91.13±0.32 | 92.04±0.49 | 96.05±0.10 | 94.78±0.24 |
| CMA-ES (AGN+Train, $\alpha$=0.05) | 93.10±0.08 | 93.62±0.08 | 95.52±0.09 | 95.84±0.13 |
| CMA-ES (AGN+Train, $\alpha$=0.01) | 93.38±0.18 | 93.39±0.06 | 95.67±0.02 | 94.39±1.42 |
| CMA-ES (AGN+Train, $\alpha$=0.005) | 92.61±0.39 | 93.39±0.20 | 95.48±0.03 | 94.51±1.45 |
| CMA-ES (AGN+Val, $\alpha$=0.1) | 93.27±0.04 | 93.43±0.08 | 96.24±0.04 | 95.59±0.05 |
| CMA-ES (AGN+Val, $\alpha$=0.05) | 93.42±0.16 | 93.45±0.17 | 95.63±0.02 | 95.96±0.04 |
| CMA-ES (AGN+Val, $\alpha$=0.005) | 93.19±0.19 | 93.60±0.05 | 95.97±0.13 | 96.13±0.07 |
| PSO (AGN+Train, $\alpha$=0.05) | 93.29±0.28 | 92.96±0.10 | 91.87±0.24 | 95.70±0.17 |
| PSO (AGN+Train, $\alpha$=0.01) | 93.21±0.06 | 92.90±0.03 | 95.79±0.13 | 96.04±0.02 |
| PSO (AGN+Train, $\alpha$=0.005) | 93.36±0.11 | 92.95±0.03 | 95.55±0.24 | 96.14±0.02 |
| PSO (AGN+Val, $\alpha$=0.1) | **93.91±0.04** | 93.82±0.08 | 95.33±0.13 | 93.90±0.30 |
| PSO (AGN+Val, $\alpha$=0.05) | 92.35±0.11 | 93.46±0.07 | 96.10±0.12 | 95.78±0.11 |
| PSO (AGN+Val, $\alpha$=0.005) | 93.09±0.10 | **93.82±0.01** | 95.52±0.07 | **96.39±0.13** |

gap demonstrates the effectiveness of using AGN to complement hyper-parameter optimization algorithm.

**AGN Helps Avoid Overfitting the Validation Set**   An interesting observation is that using the method HS1 + Val to search for hyper-parameters yields worse test set performance than the model trained with default hyper-parameters. This is likely because the model is able to overfit the validation set after multiple rounds of training during hyper-parameter optimization. Consequently, the model cannot generalize well on the test set. In contrast, when AGN is used in the metric, we can further improve the baseline model with default hyper-parameters by $0.21\%$ (the test performance using default hyper-parameter is $93.34 \pm 0.07$ whereas using CMA-ES(AGN+Train,$\alpha$=0.1) is $93.51 \pm 0.19$.

**AGN Helps Hyper-parameter Optimization Algorithm Find More Stable Settings**   When compared to Fig. 4b and Fig. 4c, Fig. 4d shows that the incorporation of AGN enables the search algorithm to find a set of hyper-parameters that gives rise to better and more stable performance on the test set as the model is being trained. The magnitude of test loss in both RS(Val) and CMA-ES(Val) can be 10 times bigger than training loss or even worse, whereas CMA-ES(AGN+Train) finds a set of hyper-parameters that keeps test loss relatively close to training loss throughout. The test loss of method CMA-ES(Val) can be as high as 7 in the midst of training. However, the loss curve is far more smooth in the case of CMA-ES(AGN+Train), as shown in Fig. 4d.

