# OpenReview forum: "Can We Use Gradient Norm as a Measure of Generalization Error for Model Selection in Practice?"
_ICLR.cc/2021/Conference — Reject_

### Official Review · AnonReviewer2 · 2020-10-26
**Can We Use Gradient Norm as a Measure of Generalization Error for Model Selection in Practice**

**Rating:** 6
**Confidence:** 3

**Review:**

In this paper, they provide the empirical studies  to understand the effectiveness and efficiency of the use of the gradient norm (induced by [the Li et al., 2020]) as the model selection criterion. To speed up the calculation process the of the gradient norm, they first propose an approximate gradient norm (AGN) based on the depth-wise, sample-wise and epoch-wise accelerations.  Their empirical studies find that the use of AGN can select the models with lower generalization error, but fails for bandit-based or population-based algorithms, and fails to predict the generalization performance of models based on different architectures.  In conclusion, they do not recommend using (approxiamte) gradient norm for model selection in practice.

Pros:

-1:  In this paper, they propose an approximate gradient norm (AGN) based on an accelerated approximation (Goodfellow, 2015) of gradient norm that only computes the loss gradient in the Fully-Connected Layers, which can significantly reduce the computation cost (200∼20,000 times faster) than the original one. They also find in empirical evaluations that AGN and GN behave identically with respect to empirical generalization gap.

-2: They carry out extensive experiments to validate the correlations between generalization performance and AGN, and find that, when the models are well-fitted, AGN well corresponds with the empirical generalization gap, but for the  under-fitted models, the correlations  between empirical generalization gap and AGN are not consistent.

-3: They use AGN  as an objective for  hyper-parameter selection under both blackbox optimization and bandit-based search, and find that,  AGN may help black-box optimization algorithms with an additional hyper-parameter, but fails to bandit-based or population-based algorithms, and fails to predict the generalization performance of models based on different architectures.

Cons:

-1: In Figure 1(a) and (c), we can find that for  Multi-Layer Preceptors (MLP), the trends of AGN behave identically with respect to the generalization gap, but, in Figure 3, they show that for ResNet-20, ResNet-56, ResNet-110 and DenseNet 100*24, the correlations between AGN and generalization gap are not consistent. These results confuse me, can we draw a new conclusion that, when the model is simple,  the  correlations  between AGN and generalization gap are consistent, but for complex models, the correlations are not consistent.  I want to see the additional experiments on the analysis of the MLP in Figure 3, I am surprised why ignoring the MLP in Section 3 and Section 4.

-2:  In Figure 1(e) and (f), one can see that the Pearson coeff  for epoch 81-120 are higher than that of the Epoch 1-80 and Epoch 121-160. This result seems a bit counterintuitive. Why the middle epoch can obtain the highest Pearson coeff?

---

> ### Author Response · Authors · 2020-11-17
> **Thank you for your comments**
>
> Many thanks for your constructive comments. We believe all your comments could greatly help us improve the manuscript. We are now working hard in revising the manuscript to address your concerns. The updated version will be available shortly. Here goes a quick response to your comments. We hope our response clarifies your questions. We hope the manuscript still can receive your full consideration for acceptance.
>
> 1.	We realize that the plots using MLP on MNIST are not the most representative of the correlation between GN/AGN and generalization gap because MNIST could be easily overfitted after training 40 epochs. We still refer to Fig. 3 in the manuscript as our main empirical result for the correlation between AGN and generalization in practice. We will update the manuscript to address your concern. The updated version will be made available soon.
> 2.	In Fig. 1(e) and (f), the differences in Pearson coefficients are mainly due to the varying scales of the plots. The Pearson coefficients are the lowest in the first 80 epochs because there are points in the early optimization path that have significantly larger values for both GN and AGN. The existence of several such extreme values causes the scale to be wide in the leftmost plot, which in turn causes Pearson coefficients to be lower. In contrast, the Pearson coefficients are both nearly 1 for epochs 80-120 and 121-160 because both GN and AGN tend to stabilize around a small value as the model gets closer to convergence.
>
> Please feel free to comment on the thread of discussion and timely shepherd us for improving the manuscript.

---

### Official Review · AnonReviewer3 · 2020-10-27
**Sum of Gradient Norms as Measure for Generalization**

**Rating:** 4
**Confidence:** 4

**Review:**

The paper empirically investigates the sum of gradient norms as a measure to determine the generalization abilities of a neural network. The approach is inspired by the theoretical work of Li, et al. 2020 which showed that the generalization gap can be upper bounded by a function of the sum of the full gradient norms of the training path.

The paper empirically investigates the gradient norm as a measure to predict the generalization gap of neural networks. For that, the paper proposes an approximation of the sum of the full gradient norms that is more computationally efficient. The paper shows that this approximation is, albeit being in a different scale, strongly correlated with the full gradient measure. The paper then continues to openly investigate the properties of the measure and finds that it is only partially correlated with the generalization gap. When used for hyperparameter tuning, it can slightly improve the results. Lastly, the paper shows that the measure is not effective in predicting the generalization ability of DNNs with different architectures.
The empirical evaluation is sound (although limited to only fairly simple image classification problems), the results are interesting, and I particularly regard the honesty about the negative results. I think that this is a valuable study.

What is missing though is a discussion of these findings and a placement of these results into the wider context of generalization in deep learning. Do these results imply that the gradient norm is not a sufficient measurement? Or can the gradient norm be an effective measure in certain situations? The paper hints at that, saying that for well-trained networks the measure correlates more strongly with the generalization gap.

A particularly interesting discussion here would be the relation to flatness measures which have been found to be strongly correlated with the generalization gap [1,2,3,4,5]. It seems, the gradient path norm does not necessarily outperform those measures but might be related. I.e., if a model is in a very flat minimum, at least for a part of its training path (the last part) it will have been in that flat region and thus gradient norms would be small.

In summary, the paper presents an interesting empirical analysis and makes some technical contributions in the form of the proposed AGN measure. However, the missing in-depth discussion limits the contribution of this paper. Since such a discussion would go beyond the minor edits for a CRC, I am not convinced this paper is ready for publication, yet.

More out of interest than as a critique of this paper, I would like the authors to clarify how the gradient norm could be a meaningful measure to determine generalization in general. Li, et al. 2020 have shown a modified PAC-Bayesian bound that uses the gradient norm over the path, but it seems that most of the heavy lifting there is done by assuming a distribution dependent (i.e., perfect) prior - please correct me if I am wrong here. It seems to me then that the gradient norm can at best be a part of the explanation for good generalization (i.e., the one that relates to flatness and thus robustness). To illustrate my point, assume your data is drawn with x uniform in R and y = cos(v*x) + eps, where eps is some Gaussian noise. That is, the data is a noisy cosine function. Our model class consists of functions f_w (x) = cos(w*x) with a single parameter w. The optimal model has parameter w^*=v. Now the error surface of that model space if we use, for example, the squared loss, has lots of local minima of various depths and one global minimum at w^*=v (both in terms of empirical risk and true risk). If we now initialize our model randomly, it can start at a steep part of a local minimum and jump into the next and from there on to the next with fairly large gradients until it ends up (hopefully) in the global minium, where it will not escape (all also depending on the learning rate, of course). In this hypothetical (and arguably quite artificial) example, the path from most initializations to the global minimum would have large gradient norms. If we instead initialize close to a local minimum, the gradients will be small and if we start close enough to the minimum, the model will not escape. It will remain in that locally flat area with very small gradients. Thus, the gradient norm of the path is very small, yet the generalization error is large (the gap then depends on the actual sample and can be either small or quite large). My question now is: am I misunderstanding Li et al. or is my example too simple? Could the authors give some intuition on how the gradient norm could explain generalization?


[1] Jiang, et al. Fantastic generalization measures and where to find them. ICLR 2020.
[2] Keskar, et al. On large-batch training for deep learning: Generalization gap and sharp minima. ICLR, 2017.
[3] Petzka, et al. Relative Flatness and Generalization in the Interpolation Regime. arxiv prerpint, 2020.
[4] Neyshabur, et al. Exploring generalization in deep learning. NIPS, 2017.
[5] Tsuzuku, et al. Normalized flat minima: Exploring scale invariant definition of flat minima for neural networks using PAC-Bayesian analysis. arxive preprint, 2019.


------ After Discussion -------
I had an interesting discussion with the authors that clarified some important points. I came to the conclusion that the topic is interesting and the study is valuable, however, not yet conclusive. Consequently, the paper is not ready for publication, yet. Thus I have lowered my score by one. I hope the authors continue this work, since I'm convinced a complete empirical study on the impact of the gradient norm along the training path is insightful and a valuable contribution to the community.

---

> ### Author Response · Authors · 2020-11-17
> **Thank you for your comments**
>
> Many thanks for your constructive comments. We believe all your comments could greatly help us improve the manuscript. We are now working hard in revising the manuscript to address your concerns. The updated version will be available shortly. Here goes a quick response to your comments. We hope our response clarifies your questions. We hope the manuscript still can receive your full consideration for acceptance.
>
> We would like to summarize our conclusion about the experimental results again. GN is inherently not a worthwhile indicator of generalization due to its heavy computational cost. We thus proposed AGN as a fast and effective proxy for GN. Without considering (1) any other prior-based generalization metrics or (2) computation budget, we found that the use of AGN can help search the hyper-parameters with close performance to the ones searched by validation set. Even with validation set included in the search, AGN+Val could be slightly better than the one using validation set only. All the above experiments are based on blackbox optimization of hyper-parameters (shown in Table 1). In terms of bandit-based search which reduces computational cost, AGN did not demonstrate any advantages, no matter whether a validation set is used. After all, we conclude that when computational budget is not a concern, AGN has the potential to complement the validation set but is still not comparable to other prior-based metrics.
>
> To put GN/AGN in a wider context, we agree that GN/AGN does not necessarily outperform other state-of-the-art metrics such as the flatness of local minimum where the model converges to. In fact, we have compared the effectiveness of AGN with a few other generalization metrics in section 4.2. The comparison results demonstrate the incompetence of AGN compared to flatness metric.
>
> We note your final question in the comment. Unfortunately, it is out of our ability to answer the question though it is very interesting. This question is more relevant to [4] rather than ours.
>
> Please feel free to comment on the thread of discussion and timely shepherd us for improving the manuscript.
>
> [4] Jian Li, Xuanyuan Luo, and Mingda Qiao. On generalization error bounds of noisy gradient methods for non-convex learning. In International Conference on Learning Representations, 2020.

---

> > ### Comment · AnonReviewer3 · 2020-11-20
> > **Thanks for your reply**
> >
> > Thank you for your reply. If I understand it correctly, then AGN is not useful in practice: it improves parameter evaluation at the cost of runtime, but is not competitive with other metrics. At the same time, it remains unclear whether AGN or GN could even be a sound metric. Thus, I am unsure what the contribution of this paper then is. Maybe I am misunderstanding you, though. Thus, may I ask you to elaborate on your contribution and why it is significant?

---

> > > ### Author Response · Authors · 2020-11-20
> > > **Our contributions and their significance**
> > >
> > > Thank you very much for your follow-up question. Our contribution is re-iterated as follows. There are previously theoretical works that demonstrate gradient norm as a measure of generalization [4, 5]. Although they have extensive and convincing theoretical proofs that show GN is a sound generalization metric, there is a lack of empirical study to verify (1) whether the metric behaves as expected in practice and (2) whether it could be employed as a generalization metric efficiently in practice. Our work fills this gap by
> > > 1.	Proposing AGN, a fast and effective proxy that makes it possible to use GN in practice (main results are shown in Fig. 1 and 2 in the manuscript).
> > > 2.	Conducting extensive experiments to validate in what cases the metric performs well and when it does not behave as expected (shown in Fig. 3 in the manuscript).
> > > 3.	Evaluating the strengths and shortcomings of using GN/AGN in practice in the context of hyper-parameter search and cross-architectural model selection (shown in Table 1, 2, and 3 in the manuscript).
> > >
> > > We believe the above contributions are significant because
> > > 1.	The proposed approximation for GN reveals an interesting correlation between the FC layer gradient norm and the full network gradient norm. This observation might spark further interests to explore a theoretical understanding that explains the relations of gradients across different layers of a neural network model.
> > > 2.	We provide empirical verification that is not offered in the previous theoretical papers [4, 5] about gradient norm. We confirm that GN is a good metric which correlates to generalization gap in a wide range of experiments when models are well-fitted. We also observe that, when models are under-fitted, the metric is not applicable. This is a surprising observation to some people because of the opposing trend observed. As a result, it might motivate researchers to generate novel insights into gradient norm as a generalization metric.
> > > 3.	We demonstrate that when runtime is not a concern and without considering other state-of-the-art generalization measures, AGN/GN could be used in two different scenarios of model selection (i.e. hyper-parameter search for a fixed architecture and cross-architectural model selection). However, the computational overhead is heavy, causing it to be an inefficient metric in practice. Our empirical study helps practitioners avoid repeating such resource-consuming experiments. Furthermore, these limitations would motivate researchers to improve the current gradient metric with the hope to discover a more efficient one.
> > >
> > > In essence, our work reveals the fact that a sound theory might not be as appealing as it is imagined to be in practice. We strongly believe realizing the insufficiency of a theoretical result when putting into practical use is also a step forward in our community. Moreover, our findings would stir new research directions. We hope you agree with us regarding these significances. We will address your concern and emphasize the significance of our contributions in the revised manuscript.
> > >
> > > If you have more comment, please do not hesitate to post here. We will be happy to have further discussion with you. Thank you for your time.
> > >
> > > [4] Jian Li, Xuanyuan Luo, and Mingda Qiao. On generalization error bounds of noisy gradient methods for non-convex learning. In International Conference on Learning Representations, 2020.
> > > [5] Mou W, Wang L, Zhai X, et al. Generalization bounds of sgld for non-convex learning: Two theoretical viewpoints. Conference on Learning Theory. 2018: 605-638.

---

### Official Review · AnonReviewer4 · 2020-10-27
**empty**

**Rating:** 4
**Confidence:** 3

**Review:**

Authors of the paper conducted many numerical experiments on the relationship between generalization and their proposal on an approximated form of gradient norms.

The motivation for their study is the article Li et al. (2020) which discovers deep relation between generalization and gradient norms. However, as far as I can understand, what the article Li et al. (2020) precisely studies is a noisy version of gradient descent (stochastic gradient Langevin dynamics), and without the noisy assumption their theoretical analysis is not valid. Based on these, I am not convinced that studying the relation between generalization and a *variant* of gradient norms with respect to the *true* SGD is a proper topic that the ML community should consider. Many experiments in the gap should be carried on.

Moreover, in 'contribution' (3) the authors make a digression discussing generalization and hyperparameter searching, and mark as one of the major contribution. The conclusion is that their 'approximated gradient norm' is not well-behaved and has many constrains in application, so I would mark these as merely observations rather than 'contributions'. I think this part is rather incomplete.

To me the article seems to be an experimental report on what has been observed during the numerical trials. It might be a good submission to workshop, but not qualify for ICLR main conference.

---

> ### Author Response · Authors · 2020-11-17
> **Thank you for your question and comments**
>
> Many thanks for your question and comments. We believe all your comments could greatly help us improve the manuscript. We are now working hard in revising the manuscript to address your concerns. The updated version will be available shortly. Here goes a quick response to your comments. We hope our response clarifies your questions. We hope the manuscript still can receive your full consideration for acceptance.
>
> We know that [4] established generalization upper bound for models trained by stochastic gradient Langevin dynamics (SGLD). We believe SGLD is a good proxy model to analyze stochastic gradient descent and a lot of assumptions. There are other works that derive theoretical bounds for non-convex learning by using SGLD as a convenient starting point for analysis [5, 6, 7]. SGLD is helpful for researchers to develop an operable theoretical bound. In practice, the applications in deep learning today tend to use SGD rather than SGLD. Our experiments have shown that when models are well-fitted, the bound remains applicable even if we use SGD.
>
> The title of our paper is “Can We Use Gradient Norm as a Measure of Generalization Error for Model Selection in Practice?” We thus adopt a pathway of systematic empirical studies, where we first analyze the correlation between generalization gap and gradient norms in an ad hoc manner. Then we present comprehensive experimental studies to investigate the feasibility of using gradient norm for hyper-parameter search in a black-box manner (through embedding the gradient norm metrics in the search objectives of commonly used hyper-parameter search frameworks). The experimental results contribute to the community as a test of theories and reveal the gap between a theoretic metric and its helpfulness in practice. GN is a good metric that correlates to generalization gap in a wide range of experiments. Further, it would be a good search objective when computational budget is not an issue. This finding would potentially motivate the community to improve gradient norm as a better and more efficient generalization metric.
>
> Please feel free to comment on the thread of discussion and timely shepherd us for improving the manuscript.
>
> [4] Jian Li, Xuanyuan Luo, and Mingda Qiao. On generalization error bounds of noisy gradient methods for non-convex learning. In International Conference on Learning Representations, 2020.
> [5] Mou W, Wang L, Zhai X, et al. Generalization bounds of sgld for non-convex learning: Two theoretical viewpoints. Conference on Learning Theory. 2018: 605-638.
> [6] Negrea J, Haghifam M, Dziugaite G K, et al. Information-Theoretic Generalization Bounds for SGLD via Data-Dependent Estimates. Advances in Neural Information Processing Systems. 2019: 11015-11025.
> [7] Pensia A, Jog V, Loh P L. Generalization error bounds for noisy, iterative algorithms[C]//2018 IEEE International Symposium on Information Theory (ISIT). IEEE, 2018: 546-550.

---

### Official Review · AnonReviewer1 · 2020-10-28
**Gradient norm is interesting to be studied, but the specific approximation proposed does not seem to be a good.**

**Rating:** 4
**Confidence:** 4

**Review:**

Summary:
This paper studies the gradient norm as a measure of generalization in deep learning. The authors first an approximation to the gradient norm (GN) that is the norm of the gradients for only fully connected layers (AGN). Then they empirically evaluate the correlation between AGN and GN as well as GN and the generalization error. In Section 2.1, the authors conclude that AGN is highly correlated with GN and both are correlated with generalization error. In Section 3, the authors conclude that the correlation between AGN and generalization error is not consistent in a wider family of models. In Section 4, authors propose to use AGN for model selection and conclude that AGN is not good for model selection unless the hyperparameter for mixing AGN with another metric is optimal.

Pros:
- Recently there has been huge interest in measures of generalization error. Studies of this sort even with mixed results can be helpful in better evaluation of individual proposed measures.
- Gradient norm is a measure proposed by prior published work (Li et al. (2020)) with theoretical justifications without extensive empirical evaluations which is done in this work.

Cons:
- The main limitation of this work is that the authors propose an approximate measure to gradient norm (AGN), then the results are mostly negative for AGN in Section 3 and 4. But this doesn’t necessarily mean gradient norm is a poor measure of generalization. The link in Section 2 is only empirically shown for a MNIST, Fashion-MNIST, and CIFAR-100 datasets and there is no theoretical reason to justify this. So I don’t find the results to be conclusive about gradient norm.
- Gradient norm can be computed efficiently in much less time than the claimed time in the paper. See [1] for a framework that computes gradient norm efficiently and [2] for general dot products between gradients.
- Empirical results in Fig 1 are the main support of the paper for using the approximate gradient norm in the next sections. It is not clear if the conclusions from MNIST, Fashion-MNIST, and CIFAR-100 datasets would hold for other datasets. Especially that the conclusion in future sections seems to be negative for the application of AGN.

Additional notes:
- In abstract given that only an approximation to gradient norm is computed the following statement cannot be made: “...gradient norm also fails to predict the generalization performance”.
- Fig 1, a-d: the norm of the gradient on MNIST should tend to zero by the end of the training as the loss tends to 0. Despite that, the range of values for AGN and GN in this figure is way above 100. In contrast, GN in Fig 1.e is below 1 which matches the expectation. Have models in a-d converged?
- The authors could also look into recent work on “robust” generalization measures [3]. AGN seems not to be a robust measure based on Figure 3. The authors could study better approximations to gradient norm and their robustness.

Typos:
- focuses -> focus
- some advanced measure -> measures
-  Thomas et al. (2019) did they derive TIC? Or only try it out?
- correlates to -> with
-  Thomas et al. (2019) is a posterior measure -> TIC is a
- our work make -> makes
- technical contributions -> contribution

[1] Dangel, F., Kunstner, F., & Hennig, P. (2019). BackPACK: Packing more into backprop. arXiv preprint arXiv:1912.10985.
[2] Faghri, F., Duvenaud, D., Fleet, D. J., & Ba, J. (2020). A Study of Gradient Variance in Deep Learning. arXiv preprint arXiv:2007.04532.
[3] Dziugaite, G. K., Drouin, A., Neal, B., Rajkumar, N., Caballero, E., Wang, L., ... & Roy, D. M. (2020). In Search of Robust Measures of Generalization. arXiv preprint arXiv:2010.11924.

============
After rebuttal:
I thank authors for their response. I share concerns with others reviewers and I highly encourage authors to consider answering questions suggested by Reviewer 3 at the end of their discussion. I believe a systematic study of gradient norm is interesting but this work does not provide a solid set of answers. As such, I'm reducing my rating to 4.

---

> ### Author Response · Authors · 2020-11-17
> **Thank you for your comments**
>
> Many thanks for your constructive comments on the manuscript. We believe all your comments could greatly help us improve the manuscript. We are now working hard in revising the manuscript to address your concerns. The updated version will be available shortly. Here goes a quick response to your comments. We hope our response clarifies your questions. We hope the manuscript still can receive your full consideration for acceptance.
>
> GN is inherently not a worthwhile indicator of generalization due to its heavy computational cost. It is not even feasible to conduct experiments on large-scale datasets such as CIFAR-10/100 using GN (even with Backpack). Motivated by this limitation, we propose AGN as an efficient approximation to GN. This approximation is shown to be very efficient in our experiments. For example, our proposed AGN (based on [3]) could be 200∼20,000 times faster than the naïve implementation of the gradient norm estimator, as illustrated in Fig. 2. In comparison, other implementations like Backpack only improves the naïve approach by a factor of 10. Actually, both BackPack[1] and AGN are derived from [3]. We believe the time consumption of BackPack is still unacceptable for estimating the sum of gradient norms over a complete optimization path. Our empirical experiments have demonstrated the viability of using AGN to approximate GN in practice. The theoretical understanding of why AGN approximates GN well is out of the scope of this paper for an empirical study. Nonetheless, this problem remains an important open direction for future research.
>
> Sorry for misleading in the manuscript. We are not arguing that GN completely fails to predict generalization performance. We meant to say the effectiveness of GN in predicting generalization across architectures is largely limited in comparison with other state-of-the-art metrics. Other generalization indicators have greater potential to be employed in practice than GN. Without considering (1) any other prior-based generalization metrics or (2) computation budget, we found that the use of AGN can help search the hyper-parameters with close performance to the ones searched by validation set. Even with validation set included in the search, AGN+Val could be slightly better than the one using validation set only. All the above experiments are based on blackbox optimization of hyper-parameters (shown in Table 1). In terms of bandit-based search which reduces computational cost, AGN did not demonstrate any advantages, no matter whether a validation set is used. After all, we conclude that when computational budget is NOT an issue at all, AGN has the potential to complement the validation set but is still not comparable to other prior-based metrics.
>
> Fig.1 (a) – (d) shows the total GN and AGN along the optimization path respectively (i.e. summation of GN or AGN in 40 epochs) whereas Fig. 1(e) and (f) show GN and AGN in every single epoch. Fig.1 (a) – (d) demonstrates the viability of using AGN as a proxy of GN by directly following equation (4) in our paper. Fig. 1(e) and (f) prove the same thing from a different point of view: if AGN well approximates GN in each epoch, the summation AGN is also a good approximation of GN along the optimization path. Despite the different representations, the plots show the same conclusion that the correlation between GN and AGN is strong, consistent, and statistically significant.
>
> Thank you very much for catching our typos. We will correct them in the updated manuscript, which will be made available soon.
>
> Please feel free to comment on the thread of discussion and timely shepherd us for improving the manuscript.
>
>
> [1] Dangel, F., Kunstner, F., & Hennig, P. (2019). BackPACK: Packing more into backprop. arXiv preprint arXiv:1912.10985.
> [2] Faghri, F., Duvenaud, D., Fleet, D. J., & Ba, J. (2020). A Study of Gradient Variance in Deep Learning. arXiv preprint arXiv:2007.04532.
> [3] Ian Goodfellow. Efficient per-example gradient computations. arXiv preprint arXiv:1510.01799, 2015.

---

### Decision · Program_Chairs · 2021-01-07
**Final Decision**

**Decision:**

Reject

**Comment:**

Dear authors,

Thank you for your submission. The reviewers all appreciated the direction of research and the message that GN can be a bad measure of generalization. That said, they all shared concerns regarding the strength of the conclusions that can be drawn from your work.

I encourage you to address their comments and submit a revised version to a later conference.

---------------------------------
Reviewer 1 wanted to update their review but couldn't so here is the update:


Some more details on my original concerns

Thank you for your detailed responses. I wanted to add more details to the ones not discussed by other reviewers.

- Regarding the speed of computing the gradient norm, I still don't agree that the computation cost is high. Figure 6 in the Backpack paper shows the cost of computing individual gradients at most 4x the cost of a single backprop not 100-1000x. In reference [2] that I gave, there is also a cheaper approximation discussed with computational costs detailed in Appendix B. As long as the computation of gradient norm is comparable with the cost of a single back-prop it should be cheap enough to run all your experiments.

- Regarding the conclusions in the paper. Thank you for giving more details. Adding those explanations to the paper would help. I personally missed some of those takeaway messages.

Overall, I strongly recommend either strengthening the link between GN and AGN or using better approximations. As well as better discussing the conclusions. Of course in addition to the suggestions by other reviewers.